



# Integrating Point Sources to Map Anthropogenic Atmospheric Mercury Emissions in China, 1978–2021

Yuying Cui[1,2], Qingru Wu[1,2*], Shuxiao Wang[1,2], Kaiyun Liu[3], Shengyue Li[1,2], Zhezhe Shi[1,2], Daiwei Ouyang[1,2], Zhongyan Li[4], Qinqin Chen[1,2], Changwei Lü[5,6], Fei Xie[5,6], Yi Tang[7], Yan Wang[8], Jiming Hao[1,2]

[1]State Key Joint Laboratory of Environment Simulation and Pollution Control, School of Environment, Tsinghua University, Beijing 100084, China
[2]State Environmental Protection Key Laboratory of Sources and Control of Air Pollution Complex, Beijing 100084, China
[3]College of Environmental Science and Engineering, North China Electric Power University, Beijing, 102206, PR China
[4]Weiyang College, Tsinghua University, Beijing 100084, China
[5]School of Ecology and Environment, Inner Mongolia University, 010021, Hohhot, China
[6]Institute of Environmental Geology, Inner Mongolia University, 010021, Hohhot, China
[7]State Key Laboratory of Environmental Criteria and Risk Assessment, Chinese Research Academy of Environmental Sciences, 100012, Beijing, China
[8]College of Electromechanical Engineering, Qingdao University of Science and Technology, Qingdao 266061, China

*Correspondence to*: Qingru Wu (qrwu@tsinghua.edu.cn)

**Abstract.** Mercury emissions from human activities persist in the environment, posing risks to humans and ecosystem, and are regulated by the Minamata Convention. Understanding the historical emissions of mercury is critical for explaining the presence of mercury in the environment. In recent years, some studies have looked at the historical trends of atmospheric emission inventory. The spatial resolution of inventories for relatively recent years have improved. However, limited inventories have combined both long time scales and high spatial resolution, which is essential for evaluating the legacy impacts of anthropogenic mercury emissions, particularly in regions with high levels of mercury emissions. Here we compile a new comprehensive point source database by fusing multiple data source, and integrate it with previous China Atmospheric Mercury Emission Model to create an annual point source and gridded emission inventory for China covering 1978-2021. Integrating point source emission inventory (P-CAME) improves the accuracy of the gridded emissions, reducing the normalized mean error for all grids by 108% compared to not using point sources in the most recent year of 2021. The improved gridded emissions inventory notably enhances the simulation of atmospheric mercury concentrations, particularly in urban areas. P-CAME inventory resulted in a 20-23% reduction in the normalized mean bias. The improved gridded emission data identifies potential polluted grids characterized by high cumulative emissions. It indicates that 20% of cumulative emissions originate from just 0.3% of the grids, primarily distributed in Gansu, Yunnan, and Hunan Provinces. These areas are predominantly dominated by non-ferrous metal smelters or a mix of emissions sources including coal-fired industries and cement production. With the improvements in simulation accuracy and the identification of highly polluted regions, this



updated inventory would greatly facilitate the assessment of mercury exposure, legacy impacts, and effective management of
cross-media mercury pollution.
**Keywords**
Speciated mercury; Emission inventory; Mercury concentration; GEOS-Chem; Cumulative emissions.



## 1 Introduction

Mercury is a persistent environmental pollutant that is harmful to the nervous systems and can affect health across generations. Human activities have liberated mercury from stable long-lived reservoirs, mainly geologic deposits and coal, to the Earth's surface, leading to 3-5 fold increase in mercury content in the land, atmosphere, and oceans since the industrial revolution (Streets et al., 2011; Corbitt et al., 2011; Selin, 2009; Selin et al., 2008). The increased load of mercury in the environment poses significantly risks to human health and ecosystem worldwide (Selin, 2009; Bishop et al., 2020; Amos et al., 2013; Li et al., 2022; Meng et al., 2011; Giang and Selin, 2016; Smith-Downey et al., 2010), promoting the establishment of the Minamata Convention on Mercury in 2013, a legally binding international treaty aimed at regulating mercury use and emissions in human activities. In accordance with the convention's regulation, the fifth Conference of the Parties had formally initiated the first effectiveness evaluation of the Convention at the end of 2023. Updated historical mercury emissions, with both temporal continuity and spatial precision, are critical and urgent to understand the changing trajectory and present state of mercury pollution and to evaluate the effectiveness of pollution control efforts.

Amidst a wide array of studies, three main global emission inventories stand out for their comprehensiveness: those established by Streets (Streets et al., 2011; Streets et al., 2019), EDGAR (Muntean et al., 2018; Muntean et al., 2014), and AMAP/UNEP (AMAP/UNEP, 2013, 2019). Nonetheless, discrepancies existed among these inventories in terms of emission quantities, species profiles, temporal trends and spatial precisions. Particularly, China has garnered great attention due to its substantial emission levels, complex source profiles, and swift advancement of control technologies. These factors collectively pose challenges to precisely estimate atmospheric mercury emissions in China. Prior researches have reduced the uncertainty of emission factors through extensive field experiments in China, culminating in the development of regional, sectoral, and national emission inventories in specific years (Wu et al., 2006; Tian et al., 2010; Tian et al., 2015; Zhang et al., 2015; Zhao et al., 2015; Wu et al., 2016; Liu et al., 2019b; Zhang et al., 2023).

Among these, three notable decadal emission inventories have been developed (Tian et al., 2015; Wu et al., 2016; Zhang et al., 2023). Yet, variations in emission trends, particularly after 2010, were pronounced. Tian et al., (2015) neither provided long-term spatial characteristics nor species profiles, limiting the comprehensiveness of their inventory. Wu et al., (2016) presented gridded emission data, but its reliance on population and GDP proxies introduced a notable degree of uncertainty regarding spatial accuracy. Zhang et al., (2023) took a step forward by aligning emissions from several critical sectors with point-source locations; however, detailed gridded emissions were made available only for 2010, 2015, and 2020. These inventories underscore a persistent gap in fine-resolution gridded and speciated mercury emission data in China, which is essential for evaluating the present state of mercury pollution and supporting effective regulatory actions.

Here we introduce a novel, speciated annual mercury emission inventories spanning from 1978 to 2021, derived from the Point-source Integrated China Atmospheric Mercury Emission Model, herein referred to as P-CAME inventories. This updated



inventory opens avenues for enhancing our comprehension of atmospheric mercury pollution. Crucially, our inventory's
accurate, annual, high-resolution emission maps can identify cumulative emission hotspots, and highlight areas of potential
multi-media environmental impacts. This inventory is publicly accessible and maintains temporal and spatial consistency with
detailed information; therefore, it lays a solid foundation for discussions on anthropogenic emissions, atmospheric pollution
and health implications. Furthermore, it is poised to offer robust support for the inaugural evaluation of the effectiveness of
the Minamata Convention.

## 2 Methods


### 2.1 P-CAME Emission inventory


This study coupled the China Atmospheric Mercury Emission Model (Zhang et al., 2015; Wu et al., 2016) with the point source
database to generate the P-CAME emission inventory. The studied 24 sectors (Table S1) were divided into 3 categories (Tier
1-3). Tier 1 was the point source emission category, including coal-fired power plant (CFPP), zinc smelting (Zn), leading
smelting (Pb), copper smelting (Cu), cement production (CEM), iron and steel production (ISP), Coal-fired industrial boilers
(CFIB), Municipal solid waste incineration (MSWI), Large scale golden production (LSGP). Emissions in Tier1 were
computed using facility-level activity and dynamic technology-based emission factors (Equation S1). Emissions from other
sectors were calculated using provincial activity data combined with probabilistic technology-based emission factors (Tier2,
Equation S2) or time-varying emission factors (Tier3, Equation S3). To acquire gridded emissions for sectors in Tier 2 and
Tier 3, source-specific spatial proxies (Table S1) were used to allocate provincial area sources to grids at a resolution of
0.25°×0.3125°. Emissions from each point source were assigned to the grid corresponding to their geographical coordinates
and combined with nonpoint source data to create comprehensive emission maps at a resolution of 0.25°×0.3125° for total
mercury ($Hg^T$) and each mercury species, namely gaseous elemental mercury ($Hg^0$), gaseous oxidized mercury ($Hg^{II}$), and
particulate-bound mercury ($Hg_P$). The new inventories, encompassing speciated mercury emissions from point sources,
nonpoint sources, were named as P-CAME. Annual emission inventories for each mercury species during 1978-2021 are
available.

### 2.1.1 Point source emission model (Tier 1)


Annual facility-level activity was from point source database. Point source database combined point sources we could get from
Environmental Statistics, Industry Associations, Pollution Source Censuses, yearbooks of various industry sectors and
previous studies, as shown in Table S2. To construct the point source database, detailed data collected for each facility included
corporate name, type of industry, capacity, types of raw materials or fuels, production or consumption levels, production or
combustion processes, control technologies, and geographical information. Data from various sources were integrated based
on the Unified Corporate Social Credit Code unique to each enterprise. Missing information of point sources in the database



were addressed using data retrieval or assimilation methods. The Baidu Map System (http://jingweidu.757dy.com/) and
Qichacha website (https://www.qcc.com/) were used to fill in missing coordinates and operational years, respectively. For
2013-2021, we acquired point sources activity for each year and validated and adjusted the activity by comparing it with
provincial activity from the yearbook. For earlier years, where varying activity were more difficult to obtain, we used point
source data from the best-validated year and time-varying provincial activity to estimate point source activity for the period
1978-2012. Specifically, for the annual activity, we first extracted data of operating facilities in the current year based on their
operational years. Then, the activity was obtained by multiplying the provincial activity in that year by the proportion of the
point source activity in the province. Dynamic technology-based emission factors for point sources were derived from
provincial mercury concentrations in fuel or raw materials, combustion or production technology release rates, air pollution
control device (APCDs) removal efficiencies, and speciation profiles (Equation S1). Raw mercury concentrations in fuel or
raw materials were obtained from our previous studies (Zhang et al., 2012; Wu et al., 2012; Liu et al., 2018). Release rates and
mercury removal efficiency were from field experiments (Zhang et al., 2016; Zhang, 2012; Chang and Ghorishi, 2003; Omine
et al., 2012). The removal efficiencies and speciation profiles for APCDs were detailed in Table S3.

### 2.1.2 Probabilistic technology-based emission model (Tier 2)

Annual provincial activity for sectors in Tier 2 were obtained from statistical yearbooks (Table S2). Probabilistic technology-
based emission factors were calculated by the provincial mercury concentration in fuel or raw materials, the release rate
associated with combustion or production technology, the removal efficiency of air pollution control devices (APCDs), the
proportion of mercury species (Equation S2). Raw mercury concentration data in fuel or raw materials were sourced from our
previous studies (Zhang et al., 2012; Wu et al., 2012; Liu et al., 2018; Liu et al., 2019b). The release rate was determined based
on the specifications outlined in the preceding section for coal-fired sectors. The release rates and mercury removal efficiency
were from field experiments (Zhang et al., 2016; Zhang, 2012; Chang and Ghorishi, 2003; Omine et al., 2012). The removal
efficiency and the proportion of speciated mercury in different APCDs combinations was derived from our previous studies
(Liu et al., 2019b; Zhang et al., 2023; Wu et al., 2016).

### 2.1.3 Time-varying emission model (Tier 3)

Annual provincial activity for sectors in Tier 3 were obtained from statistical yearbooks, Chinese environmental statistics, and
investigation reports (Table S2). The emission factors for Tier 3 sectors dynamically changed with technology iterations,
assuming that emission factors fit a transformed normal distribution due to the dynamics of technology change (Tian et al.,
2015; Streets et al., 2011). The emission factor for a specific year was calculated using the emission factor at the beginning
year of technology transition ($ef_a$) and the best achievable emission factor ($ef_b$), as outlined in Equation S3. The parameters $ef_a$,
$ef_b$, and the curve shape parameter S were derived from previous studies (Tian et al., 2015; Streets et al., 2011; Wu et al., 2016;
Wu et al., 2006; Zhang et al., 2015).



Provincial emissions from sectors in Tier 2 and Tier 3 were allocated to grids at a resolution of 0.25°×0.3125° using a newly
developed spatial allocation system, as detailed in Table S1. This allocation relied on proxies such as GDP, population data,
and a roadmap dataset. Provincial non-point sources were first allocated to the city level based on GDP and then further
distributed to the grid level using either population or road network datasets, as specified in Equation S4. City-level GDP data
were extracted from statistical yearbooks, with the GDPs of primary, secondary, and tertiary industries utilized for various
sectors, as detailed in Table S1. Population data at the grid level were obtained from the resource and environmental science
data registration and publication system (Xu, 2017). While population data were available for select years (1990, 1995, 2000,
2005, 2010, 2015, and 2019), data for intermediate years were interpolated. Specifically, data for the years 1978-1989 were
estimated based on available data and observed trends during the period of 1990-2000. Road network data utilized in this study
were sourced from OpenStreetMap (https://www.openstreetmap.org/). The widths of various route types in the road network
were determined based on classifications provided in the Interim Provisions on Urban Planning Quota Index (MOHURD,
1980). These routes were then converted into areas and subdivided into grids. The gridded routes served as a proxy for the
spatial distribution of atmospheric mercury emissions from the transportation sector. To develop this long-term anthropogenic
mercury emission dataset, software tools such as ArcGIS and Matlab were employed.

## 2.2 Uncertainty analysis

Monte Carlo simulation assessed mercury emission uncertainty using key parameters and their probability distributions.
Parameters included activities, mercury concentrations in fuel/raw materials, and mercury removal efficiencies of APCDs.
Activities were normally distributed with variation coefficients of 5%-30% (Liu et al., 2019a). Mercury concentrations
followed a log-normal distribution and mercury removal efficiencies followed normal or Weibull distributions, which were
generated based on field experiments (Zhang et al., 2012; Wu et al., 2012; Liu et al., 2018; Zhang et al., 2016; Zhang, 2012;
Chang and Ghorishi, 2003; Omine et al., 2012). MATLAB conducted 10,000 Monte Carlo simulations. Mean values served
as best estimates, with 2.5% and 97.5% quantiles establishing lower and upper limits of simulation results.

## 2.3 Simulation and validation comparison

To assess the impacts of the point source inventory, we designed two simulation scenarios with different anthropogenic
emissions inputs: one using P-CAME and the other relying solely on proxies for emission allocation, referred to as "only
proxy-based" thereafter. In the only proxy-based inventory, original point source sectors were initially calculated at the
provincial level and then distributed to grids based on secondary GDP and population. Then we compared the simulations
from both scenarios with observations of atmospheric mercury concentrations. Observation data in 6 sites including 3 urban
observation sites and 3 rural observation sites (Sun et al., 2024; Wu et al., 2023; Shao et al., 2022) were collected in this study,
as detailed in Table S4. Atmospheric mercury concentrations were simulated using a global 3-D atmospheric chemistry model
(GEOS-Chem, v12.6.3, http://geos-chem.org) at a resolution of 0.25°×0.3125°. Considering the availability of observational





data, we ran nested simulation in China at 2021, with a three-year spin-up (2018-2020) was conducted for initialization.
Meteorological data were driven by GEOS Forward Processing meteorological data (GEOS-FP). Boundary conditions were
obtained from global simulations at a resolution of 2.0° × 2.5°. In addition to our anthropogenic emissions inventory, emissions
data included natural emissions from geogenic activities, biomass burning, soil, and ocean, as configured in GEOS-Chem
based on the methodology outlined by Selin et al., (2008).

## 3 Results

### 3.1 Spatial distribution pattern of atmospheric mercury emissions

This study developed an extensive point source database covering the period from 1978 to 2021. For instance, in the most
recent year of 2021, the inventory includes over 26,000 industrial facilities. Atmospheric mercury emissions in 2021 were
estimated to be 358 t, with $Hg^0$, $Hg^{II}$, and $Hg_P$ accounting for 55%, 43%, and 2%, respectively. The point source emissions
accounted for over 85% in 2021. The point sources were unevenly distributed, primarily concentrated in East and South China
(Fig. 1). Their emissions exhibited a broad spectrum of orders of magnitude, with 90% of the total emissions budget being
dominated by only the top one third large point sources (Fig. 1). Among the top one third large point sources, 68% were cement
production (CEM) facilities, widely distributed in North China, East China, South China, Central China and Southwest China
as indicated by P-CAME.

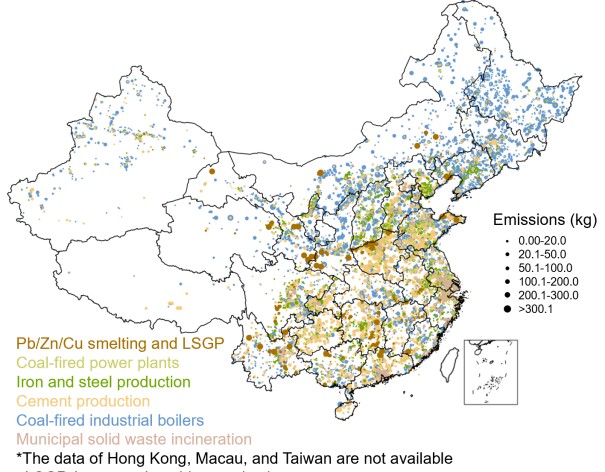

**Figure 1 Spatial distribution of point source emissions at 2021.**

The integration of point sources in the P-CAME inventories improved the accuracy of the located gridded emissions, compared
to the only proxy-based inventory (Fig. 2a & b). To quantify these differences, the normalized mean bias (NMB, Equation S1)
and the normalized mean error (NME, Equation S2) were employed. The calculated NMB and NME for all grids stood at 1%





and 108%, respectively. The low NMB alongside the high NME indicated a pronounced discrepancy between P-CAME
inventories and the only proxy-based inventory, primarily due to misalignment in grids with high and low emissions. Overall,
proxy method tented to overestimate emissions in densely populated areas, notably in capital cities such as Lanzhou, Xi'an,
Kunming, Guizhou and Guangdong, while significantly underestimated emissions in industrial clusters like Jiaozuo, Baoji,
Handan, Tangshan and Chenzhou (Fig. 2c). At a more granular grid scale, discrepancies included both overestimations and
underestimations. For example, in Handan's grids, emissions using proxy method were overestimated in the eastern parts and
underestimated in the west, contributing to the substantial NME value (108%). This illustrated that the emission using the
proxy method inaccurately distributed emissions not just between cities but also within individual city grids, causing significant
variations.

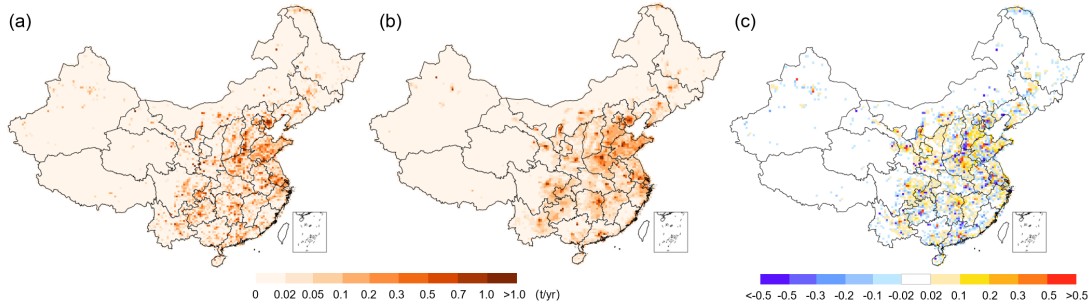


**Figure 2 Comparison of spatial distribution between (a) P-CAME and (b) the only proxy-based inventory; (c) absolute difference**
**of these two distributions.**
**3.2 Temporal trends of annual emissions**
The analysis of long-term point source emissions enabled a reassessment of historical mercury emission trends and sector
contributions from 1978 to 2021. The overall trend showcased an initial rise in emissions, peaking in 580 t, following which
the emissions declined. This trend reflecting substantial shifts across key sectors such as coal-fired power plants (CFPP), non-
ferrous metal smelting (NFMS), CEM, and coal-fired industrial boilers (CFIB) (Fig. 3a). By 1990, emissions nearly doubled
from 1978, reaching 267 t, with an average annual rising rate of 5% and CFIB, NFMS, and CFPP being the primary sources.
The NFMS emissions peaked in 2004, following which the emissions declined, while the CEM emissions rose faster, and
CEM becoming the second-largest contributor by 2010. The decade ending in 2010 saw emissions reaching 555 t, with an
average growth rate of 4%, despite a brief period of reduction due to drops in the CFPP and NFMS emissions. The following
decade highlighted a general decline in emissions from NFMS, CFIB, and CFPP, but the CEM emissions were still increasing,
making it the largest contributor to the total emissions since 2011. It was until 2021 that a slight increase in total emissions
was noted, driven mainly by rises in municipal solid waste incineration (MSWI) and the CEM emissions.

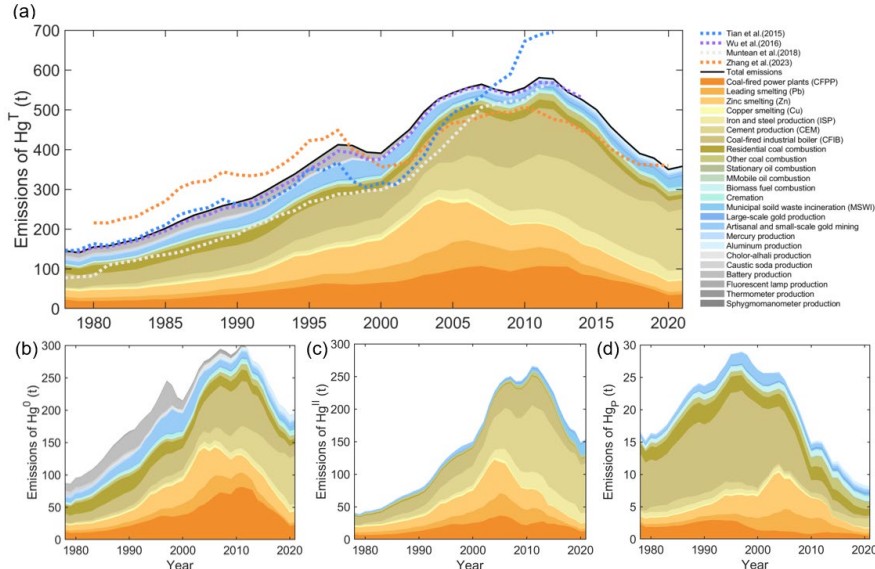

**Figure 3 Annual anthropogenic mercury emissions and comparison with other emission inventories. (a) Hg$^T$; (b) Hg$^0$; (c) Hg$^{II}$; (d) Hg$_P$.**

In line with the trend observed in total mercury emissions, annual speciated mercury emissions also followed a pattern of initial increase followed by a decline. Specifically, annual Hg$^0$ emissions rose from 87 t to 302 t during the period of 1978-2011, subsequently decreasing to 195 t by 2021 (Fig. 3b). Notably, three peaks occurred during the increasing phase of Hg$^0$ emissions: the first peak in 1997 due to battery production emissions, the second peak in 2007 resulting from reduced activity levels and enhanced SO$_2$ control in CFPP, and the third peak in 2011 due to enhanced NO$_x$ control in CFPP. Annual Hg$^{II}$ emissions increased from 41 t to 266 t during 1978-2011, followed by a decline to 155 t by 2021 (Fig. 3c). During the increasing phase of Hg$^{II}$ emissions, two peaks occurred: the first peak in 2007 was attributed to a rapid decline in NFMS and CFPP emissions during 2007-2009, while the second peak in 2011 was caused by a peak in continuous CEM emissions. Annual Hg$_P$ emissions rose from 17 t to 29 t during 1978-1997, then decreased to 8 t by 2021 (Fig. 3d), with the peak occurring in 1997 mainly dominated by emissions in CFIB.

Overall, mercury emissions in China have experienced three distinct phases: an increase from 1978 to 2007, stabilization from 2008 to 2012, and a decrease from 2013 onwards. These phases reflect varying emission and control characteristics. The first phase (1978-2007) was marked by rapid growth in activity levels, leading to a significant increase and peak in emissions. The second phase (2008-2012) saw the combined effects of continued growth in activity levels and the implementation of emission controls, resulting in relatively stable changes. The third phase (2013-2021) was characterized by a reduction in emissions driven by more stringent emission controls. These three phases were also clearly delineated in the patterns of gridded emissions depicted across three rows in Fig. S1. During the first period, there was an average 5% increase in annual emissions,



particularly noticeable in the border areas of North China, Central China, and the Yangtze River Delta (first row of Fig. S1).
Throughout the second period (2008-2012), the emissions remained relatively unchanged, with an average 0.5% increase in
annual emissions (second row of Fig. S1). In the subsequent third period (2013-2021), a noticeable reduction with an average
5% decrease in annual emissions was observed, particularly in area increased during the growth period (third row of Fig. S1).
**3.3 Comparison with previous emissions inventories**
The P-CAME emission inventory was evaluated against prior long-term inventories in China, demonstrating good alignment
with our earlier findings reported by Wu et al., (2016) and closely matching the estimates by Tian et al., (2015) until 1995 (Fig.
3a). A detailed sectoral comparison (Fig. S2) revealed that the congruence with Tian et al., (2015) was somewhat coincidental.
This study reported lower emissions from the NFMS and intentional mercury use, but higher emissions from mercury
production than Tian et al., (2015) before 1995. Post-1995, the primary discrepancies with prior studies stemmed from the
zinc, lead, copper sectors, and the CFPP. Differences with the Zhang et al., (2023) study was in two periods—before and after
1998—based on total mercury emissions (Fig. 3a). Before 1998, our study reported lower emissions, mainly attributed to a
reduced estimate from the CFIB by approximately 40 t. A higher reported utilization of air pollution control devices (APCDs)
accounted for the underestimation. After 1998, our study reported higher emissions, particularly in the CEM and NFMS sectors,
attributed to differences in mercury concentration in fuels or raw materials and the application of APCDs. The uncertainty of
P-CAME emission inventory was subjected to (-16.1%, 15.9%) in 2021, but reached (-21.8%, 21.5%) in 1978 due to the higher
uncertainty of parameters after data fusion (Fig. S3). The uncertainty ranges were among the lowest reported in existing studies
(Wu et al., 2016; Liu et al., 2019b; Zhang et al., 2023).
**4 Discussions**
**4.1 Impacts on the simulation of atmospheric mercury concentrations**
Comparisons between both simulation scenarios and observations were shown in Fig. 4. For each site, we compared monthly
average concentrations and evaluated them using NME and NMB, as shown in Table S4. Our findings indicated that P-CAME
inventory significantly enhanced GEM simulation in urban areas when compared to non-point source layers. For instance, at
the observation site in Hohhot, simulations incorporating point sources resulted in a 23% reduction in both NMB and NME
when compared to simulations without point sources (Fig. 4 & Table S4). At the Nanjing site, NMB and NME decreased by
20% and 15%, respectively (Fig. 4 & Table S4). However, the calibration results from the three rural observation stations
revealed no significant difference in NMB and MNE between simulations with and without point sources (Fig. 4 & Table S4).
The improved calibration of urban stations was primarily attributed to the use of point source emissions, which enables more
accurate localization of emissions sources. This approach mitigated the issue of overestimation of emissions in urban areas
due to high population density in emission allocation using the proxy method.





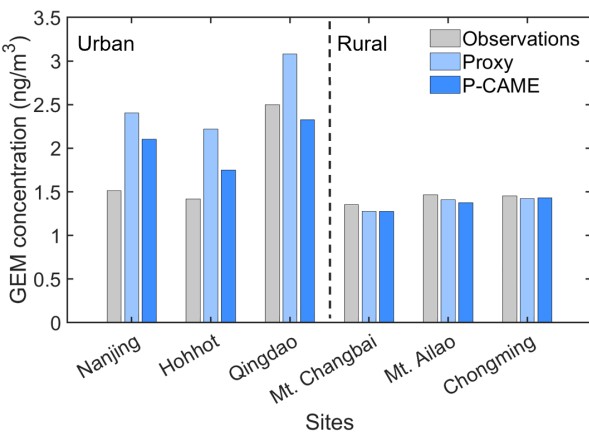


**Figure 4 Comparative analysis of observed and simulated atmospheric mercury concentrations using Proxy and P-CAME.**
**4.2 Identifying cumulative emission hotspots**
Atmospheric mercury emissions can affect human health through air inhalation; however, their deposition on surfaces and
prolonged retention pose even greater risks by causing cross-media impacts and persistent threats. The continuous, high-
resolution, and spatially detailed P-CAME inventories enable the identification of hotspots for cumulative atmospheric
mercury emissions since 1978, marking the start of China's economic expansion with its reform and opening-up policy. Over
this period, total mercury emissions reached 16,422 t, with $Hg^0$ accounting for 9,074 tons (55.3%), $Hg^{II}$ for 6,478 t (39.4%),
and $Hg_P$ for 869 t (5.3%). The cumulative emissions map, as depicted in Fig. 5a, identifies critical hotspots that, despite
covering only 0.3% of the grids, contributed to 20% of the total emissions. These hotspots, where cumulative emissions
exceeded 44 t (averaging more than 1 t annually), were chiefly found in Gansu, Yunnan, and Hunan Provinces. Emission
sources within these hotspots fall into two primary categories based on sectoral contributions: those predominantly influenced
by NFMS and those influenced by sectors other than NFMS, as shown in Fig.5b. Grids dominated by NFMS represented 76%
of the areas with high cumulative emissions, where NFMS's contributions averaged 96%. These areas also exhibited a
significant presence of $Hg^{II}$ and $Hg_P$, averaging 51%, as indicated in Fig. 5c. Conversely, grids primarily affected by other
sectors—such as CFPP, CFIB, CEM, Iron and steel production (ISP)—were located in Hebei, Henan, Hubei, Jiangsu, and
Shanghai. The sectoral contribution to the hotspots of cumulative emissions indicated that grids with NFMS tends to cause
severe cross-media mercury pollution due to their high emission intensity and $Hg^{II}$ proportion.

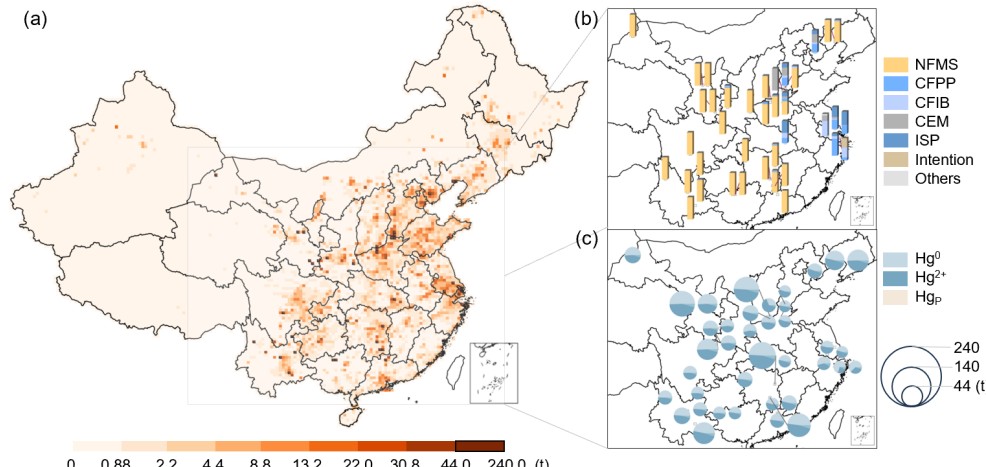


**Figure 5 Spatial distribution of cumulative mercury emissions. (a) Total mercury emissions; (b) Sectors contribution for total**

**mercury emissions in hotspots; (c) Speciation profiles in hotspots.**

To further inform future pollution control strategies, we analyzed the atmospheric mercury emission hotspots for 2021, defined
by emissions exceeding 1 t. Remarkably, half of the hotspots identified in 2021 coincided with those identified through
cumulative emission analyses (Fig. S5). These overlapping hotspots were predominantly found in Gansu, Shaanxi, Henan, and
Hebei provinces. In detail, Gansu and Shaanxi's hotspots were mainly attributed to emissions from NFMS, whereas Henan and
Hebei's hotspots were largely due to emissions from the CEM. These areas warrant heightened focus, as addressing pollution
here involved not only mitigating the impact of historical emissions but also urgently implementing controls on current
emissions to prevent further environmental degradation. Moreover, new hotspots emerging in 2021 that did not coincide with
historical cumulative emission hotspots were primarily located in Hebei, Henan, and Anhui Provinces (Fig. S5), with CEM
emissions contributing an average of 82% to these areas. While grids in Yunnan, Hunan, and Guangxi Provinces had high
cumulative emissions, their 2021 emissions did not reach similar levels. Therefore, it is clear that future efforts in pollution
prevention and control should prioritize areas with both significant cumulative emissions and high recent emissions, especially
those impacted predominantly by cement industry activities. This focused approach is essential to simultaneously tackle the
challenges of accumulated historical pollution and prevent the exacerbation of current emission levels, ensuring targeted and
effective pollution control measures.

## 5 Data availability

Integrating point source emission inventory (P-CAME) can be accessed from http://doi.org/10.6084/m9.figshare.26076907
(Cui et al., 2024).



## 6 Conclusions and implications

In this study, we introduce an annual speciated mercury emission inventories (1978-2021), P-CAME inventory. By using this novel inventory, the modelled bias of mercury concentrations in urban sites were significantly reduced, which will significantly improve the understanding of mercury cycling, and thus facilitate the assessment of potential health impacts resulting from exposure to mercury in the environment. Crucially, our inventory's accurate, annual, high-resolution emission maps can identify cumulative emission hotspots. The identification of hotspots where cumulative mercury emissions are exceptionally high suggests that targeted pollution control measures could be highly effective. By focusing on these critical areas, which contribute disproportionately to total emissions despite covering a small fraction of the land area, policymakers can allocate resources more efficiently and achieve significant reductions in overall mercury pollution. The substantial presence of $Hg^{II}$ and $Hg_P$ in areas dominated by NFMS and CEM points to the potential for severe cross-media mercury pollution. This form of pollution affects not only the air but also water bodies and soils, leading to broader environmental degradation and health risks. Strategies to mitigate mercury emissions areas such as Gansu, Shaanxi, and Hunan Provinces must therefore consider the cross-media implications of mercury pollution.

This publicly accessible inventory, characterized by its temporal and spatial consistency and detailed emission information, provides a critical foundation for nuanced discussions on anthropogenic emissions, atmospheric pollution, and their implications for human health and environmental integrity. The comprehensive nature of the data allows for a deep dive into the sources, distribution, and trends of mercury emissions, facilitating a better understanding of the global mercury cycle and identifying key areas for intervention. Moreover, the inventory's robustness and reliability are instrumental in supporting the initial evaluation of the Minamata Convention's effectiveness. As the first global treaty aimed at protecting human health and the environment from anthropogenic emissions and releases of mercury and mercury compounds, the Convention's success hinges on accurate and comprehensive data. The inventory not only aids in assessing progress towards the Convention's objectives but also highlights areas where further efforts are needed. By providing a solid empirical basis, it enables policymakers, researchers, and environmental advocates to craft more targeted and effective strategies for reducing mercury emissions, ultimately contributing to the global endeavour to mitigate atmospheric pollution and safeguard public health.

Owing to constraints in data availability, this study limited its scope to reviewing anthropogenic mercury emissions in China from 1978 onwards, with an incomplete point source coverage. To improve percentage of point sources emissions, future research can incorporate data such as satellite images and visual identity to enhance the accuracy of identification of industrial point sources, thereby refining the inventory of industrial emissions. Additionally, more studies should be conducted across multiple dimensions, including time, space, and emission impacts, potentially incorporating machine learning techniques and AI techniques to expand the temporal and spatial scope of anthropogenic emissions analysis. Those innovative methods could facilitate investigation and assessment of the long-term environmental implications of historical anthropogenic mercury emissions.





## Author contributions

Y.C. established the emission inventories and wrote the draft. Q.W. supervised the study, helped conduct data analysis, and wrote and edited the manuscript. S.W. helped conceive the idea for this article and edited the manuscript. K.L., S.L., Z.S., D.O., Z.L. helped to collect and provided basic data for calculation. Q.C. polished the draft. C.L., F.X., Y.T., Y.W. provided GEM concentration data for validation. J.H. helped conceive the idea for this article. All the co-authors revised the manuscript.

## Competing interests

The authors declare that they have no conflict of interest.

## Disclaimer

Publisher's note: Copernicus Publications remains neutral with regard to jurisdictional claims in published maps and institutional affiliations.

## Acknowledgements

We express our gratitude to the authors of the articles for providing the observation data utilized in this article. And we extend our gratitude to numerous staff members at the Environmental Protection Key Laboratory of Sources and Control of Air Pollution Complex for their invaluable contributions to supplementing the data on point sources.

## Financial support

This work was supported by the National Natural Science Foundation of China (No. 2222604, No. 42394094), and National Key Research and Development Program (No. 2022YFC3700602).

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
