# Peer review of "Integrating Point Sources to Map Anthropogenic Atmospheric Mercury Emissions in China, 1978–"

_Earth System Science Data, 2024_

## Referee Comment (RC1)

**Review Comments**

In this study, the authors established long-term Hg emission maps for anthropogenic sources in China using an integrated P-CAME model. The temporal trends and spatial distributions of sectoral Hg emissions were analyzed. Key sectors and spatial hotspots of cumulative Hg emissions were identified. This dataset could provide crucial input for chemical transport models and Hg budget models. The yields of this study are of broad interest. The manuscript is well organized and written. Overall in my opinion, the manuscript is acceptable for publication on Earth System Science Data after minor revision.

Here are some specific comments:

1. Introduction: Emission inventories are fundamental inputs for chemical transport models (CTMs). Applications of existing Hg emission inventories in CTMs and their performances in different regions can be introduced. The emission maps in this study could contribute to future atmospheric Hg simulations.

2. Section 2.1.2: The method of Monte Carlo simulation should be mentioned here instead of only in Section 2.2, with an introduction to the basic principle.

3. Section 2.3: Did the authors adopted the improvement of the GEOS-Chem model in their recent study (Liu et al., 2022)?

4. Line 183: It should be "tended" instead of "tented".

5. Line 196: It should be "reflects" instead of "reflecting".

6. Lines 239–241: What is the confidence level of the uncertainty ranges?

7. Line 251: It should be "NME" instead of "MNE".

8. Sections 3 and 4: The sub-sections in these two sections are more like parallel ones instead of results and discussion, respectively. Therefore, I recommend the authors to change the structure to a combined section "Results and Discussion". More discussion is encouraged for the current Section 3.

Reference:

Liu, K. Y., Wu, Q. R., Wang, S. X., Chang, X., Tang, Y., Wang, L., Liu, T. H., Zhang, L., Zhao, Y., Wang, Q. G., and Chen, J. S.: Improved atmospheric mercury simulation using updated gas-particle partition and organic aerosol concentrations, J. Environ. Sci., 119, 106–118, 2022.

---

## Author Comment (AC1)

**Reviewer #1**

✧ **General comments:**

In this study, the authors established long-term Hg emission maps for anthropogenic sources in China using an integrated P-CAME model. The temporal trends and spatial distributions of sectoral Hg emissions were analyzed. Key sectors and spatial hotspots of cumulative Hg emissions were identified. This dataset could provide crucial input for chemical transport models and Hg budget models. The yields of this study are of broad interest. The manuscript is well organized and written. Overall in my opinion, the manuscript is acceptable for publication on Earth System Science Data after minor revision.

**Response**: We appreciate the reviewer's comments and have addressed the detailed comments in our revised manuscript and the detailed responses below. Thank you for the comments!

✧ **Detailed comments and replies**

1. Introduction: Emission inventories are fundamental inputs for chemical transport models (CTMs). Applications of existing Hg emission inventories in CTMs and their performances in different regions can be introduced. The emission maps in this study could contribute to future atmospheric Hg simulations.

**Response**: Thank you for your comments. We have added a discussion on the applications and performance of current emission inventories in CTMs across regions and trends in the Introduction section.

"Amidst a wide array of studies, four main global emission inventories stand out for their comprehensiveness and broadly implication in CTMs (Chemical transport models): those established by Streets (Streets et al., 2011; Streets et al., 2019), EDGAR (Muntean et al., 2018; Muntean et al., 2014), AMAP/UNEP (AMAP/UNEP, 2013, 2019), and WHET (Zhang et al., 2016b). The annual emission magnitudes across inventories are ranked as WHET > Streets > AMAP/UNEP > EDGAR. Spatially, higher-emission grids are observed in WHET, Streets, and AMAP/UNEP for 2010, whereas EDGAR shows lower emissions, particularly in East and South Asia. Regarding long-term trends, EDGAR and Streets exhibit a gradual increase in emissions from 1980–2012 and 1980–2015, respectively. In contrast, WHET shows a decline followed by an increase during 1990–2010. These emission inventories have been extensively used in CTMs to simulate the atmospheric transport, transformation, and deposition of Hg. Comparing simulated $Hg^0$ concentrations with observations provides a critical metric for evaluating the performance of emission inventories in CTMs. Despite discrepancies among inventories in terms of emission magnitudes, species composition, and spatial distributions, a study employing the ECHMERIT model (Jung et al., 2009) reported no statistically significant differences in regression slopes when inventory-based simulations were compared with observational data (Simone et al., 2016). In terms of trends, both Streets and EDGAR indicate increasing emissions. However, when Streets inventory data were used as CTMs input, the simulated $Hg^0$ concentrations conflicted with the observed decline in atmospheric $Hg^0$ concentrations in the Northern Hemisphere during 2005–2020 (Feinberg et al., 2024). Anthropogenic emissions were identified as the primary driver of the divergence between simulated and observed $Hg^0$ concentrations and the associated declining trend (Feinberg et al., 2024). The WHET inventory, which incorporates updated country-specific emissions for China,

India, the U.S., and Western Europe, successfully reproduced observed atmospheric Hg concentration declines in GEOS-Chem simulations (Zhang et al., 2016b). Emission estimates from WHET for 1990, 2000, and 2010 were 1.3 to 2.4 times higher than those reported by Streets or EDGAR, highlighting the pivotal role of regional emissions in accurately capturing global emission trends and aligning them with observational data."

**See revised Manuscript, Lines 48-68.**

2. Section 2.1.2: The method of Monte Carlo simulation should be mentioned here instead of only in Section 2.2, with an introduction to the basic principle.

**Response**: Thank you for your comments. We have included an introduction to the application of Monte Carlo simulation in the calculation probabilistic technology-based emission factors in Section 2.1.2.

"To estimate mercury emissions with greater accuracy and reduced bias, Monte Carlo simulations were applied to produce probabilistic technology-based emission factors, addressing the variability and uncertainty in key parameters. Emission factors were calculated based on the provincial mercury concentration in fuel or raw materials (log-normal distribution), release rates associated with combustion or production technologies (as specified for coal-fired sectors), removal efficiencies of APCDs (normal or Weibull distributions), and the proportions of mercury species determined by APCDs combinations (Equation S2). Raw mercury concentration data and their standard deviations were sourced from previous studies (Zhang et al., 2012; Wu et al., 2012; Liu et al., 2018; Liu et al., 2019), while mercury removal efficiencies and release rates were obtained from prior research based on field experiments (Zhang et al., 2016a; Zhang, 2012; Chang and Ghorishi, 2003; Omine et al., 2012). Speciated mercury proportions for various APCD combinations were derived from our earlier work (Liu et al., 2019; Zhang et al., 2023; Wu et al., 2016). By incorporating these parameters into Monte Carlo simulations, probabilistic emission factors were generated, providing a robust and comprehensive estimation of mercury emissions across Tier 2 sectors"

**See revised Manuscript, Lines 126-137.**

3. Section 2.3: Did the authors adopted the improvement of the GEOS-Chem model in their recent study (Liu et al., 2022)?

**Response**: Thank you for your question. We did not use the improvements to the GEOS-Chem model as described in Liu et al., 2022. One significant limitation is the availability of long-term and consistent organic aerosol concentration. Additionally, our study primarily focuses on developing a gridded emission inventory by creating a long-term point source database, without incorporating these model adjustments. To clarify, we have provided further details on the GEOS-Chem mechanism used in Section 2.3.

"We applied a global 3-D atmospheric chemistry model (GEOS-Chem, v12.6.3, http://geos-chem.org) to simulate atmospheric mercury concentrations from 2006 to 2021. A three-year spin-up (2006-2008) was used to achieve balanced concentrations, which serve as the restart field for analysis year (2009-2021). The global simulation was conducted at a resolution of 2.0° × 2.5° to provide boundary conditions for a nested simulation over the China region, which had

a finer resolution of 0.5°×0.625° and 47 vertical levels. Meteorological input was driven by the Modern-Era Retrospective analysis for Research and Applications, Version 2 (MERRA2) (Gelaro et al., 2017). For the global simulation, the EDGAR emission inventory was used as it provides long-term emissions data for the entire simulation period. However, since EDGAR tends to underestimate emissions in China, we replaced China's emission with the P-CAME inventory. Biomass burning emissions were calculated based on GFED4 (van der Werf et al., 2017), while geogenic activities, soil emission and re-emission followed the calculation scheme outlined in Selin et al., (2008). The chemical scheme in v12.6.3 involves the oxidation of $Hg^0$ through a two-step mechanism initiated by Br. Photoreduction of $Hg^{2+}$ occurs in the aqueous phase and is governed by the $NO_2$ photolysis rate and organic aerosol concentrations (Horowitz et al., 2017)."

**See revised Manuscript, Lines 169-180.**

4. Line 183: It should be "tended" instead of "tented".

**Response**: Revised.

"Overall, proxy method tended to overestimate emissions in densely populated areas"

**See revised Manuscript, Line 212.**

5. Line 196: It should be "reflects" instead of "reflecting".

**Response**: Revised.

"This trend reflected substantial shifts across key sectors"

**See revised Manuscript, Line 225.**

6. Lines 239–241: What is the confidence level of the uncertainty ranges?

**Response**: The confidence level of the uncertainty ranges is 95%CI. The uncertainty range, defined by the 2.5% and 97.5% quantiles, represents a 95% confidence interval, indicating a 95% probability that the true value lies within this range. We have added this explanation to the context.

"The uncertainty range, defined by the 2.5% and 97.5% quantiles, represents a 95% confidence interval, indicating a 95% probability that the true value lies within this range. For P-CAME emission inventory, the uncertainty range was subjected to (-16.1%, to 15.9%) in 2021, reflecting lower uncertainty in the parameters."

**See revised Manuscript, Lines 269-272.**

7. Line 251: It should be "NME" instead of "MNE".

**Response**: Thank you for your comment. Since we revised this section based on feedback from another reviewer, the original content no longer exists. We have carefully reviewed the article to ensure all instances of "NME" or "NMB" are correct, and the issue is resolved.

8. Sections 3 and 4: The sub-sections in these two sections are more like parallel ones instead

of results and discussion, respectively. Therefore, I recommend the authors to change the structure to a combined section "Results and Discussion". More discussion is encouraged for the current Section 3.

**Response**: Thank you for your suggestion. We have combined Sections 3 and 4 into a single "Results and Discussion" section, as recommended, and expanded the discussion to provide more insights. This section now consists of two main parts: emissions (Section 3.1-3.4) and simulation results (Section 3.5-3.6). In the emissions part, we analyze the spatial distribution and improvements of P-CAME over the proxy method, temporal trends, comparisons with previous long-term emission inventories, and the identification of cumulative emission hotspots. In the simulation results part, we present the results of long-term simulations and evaluate the performance of P-CAME compared to the proxy method.

**See revised Manuscript, Lines 196-384.**

---

## Author Comment (AC2)

**Reviewer #2**

✧ **General comments:**

This manuscript presents a novel approach to improving the annual mercury emissions inventory for China from 1978 to 2021 using the P-CAME model. The work is important for understanding mercury emissions in the region and for supporting policy measures under the Minamata Convention. However, the validation section requires significant improvement to ensure the reliability of the data and the robustness of the conclusions. Below are some suggestions to enhance the manuscript before it can be considered for publication.

**Response**: We appreciate the reviewer's comments and have addressed them in our revised manuscript according to the detailed comments. Thank you for the comments!

✧ **Detailed comments and replies**

1. Validation over the Entire Study Period: The study covers a long time span (1978–2021), with peak emissions identified around 2010–2012, as shown in Figure 3. However, the model evaluation is limited to the year 2021, which represents a period of reduced emissions compared to the peak years. This raises concerns about whether the model performs well in earlier years, especially around the time of peak emissions. To address this issue, the authors should include validation for multiple years, particularly around periods of significant changes in emissions, such as 2010–2012. If observational data from earlier periods are scarce, the authors could explore alternative methods to compare model outputs with historical trends.

**Response**: We appreciate the reviewer's comments. We added long-term $Hg^0$ concentration simulation and comparison with observation data at 9 sites during 2009-2021. Among these sites, 6 of them included observation $Hg^0$ concentration data before 2012, which provides us chances to observe the impacts of peak emissions.

"3.5 Long-term simulation of atmospheric mercury concentrations

The temporal and spatial distributions of annual atmospheric $Hg^0$ concentration are presented in Fig. 5. During 2011 to 2021, the simulated $Hg^0$ concentrations showed a declining trend, with the maximum values decreasing from 5.7 ng/m³ to 3.0 ng/m³, and the national average dropping slightly from 1.5 ng/m³ to 1.4 ng/m³. The spatial distribution analysis (Fig. 5) highlights a decline of simulated $Hg^0$ concentration in high-emission regions. However, the simulated magnitude of decline fails to capture the observed decline at monitoring sites, primarily due to an underestimation of $Hg^0$ concentrations from 2010-2013, when anthropogenic emissions peaked in China (Fig. S5). This issue has also been existed in previous studies, which found that GEOS-Chem simulations underestimate $Hg^0$ concentration during this period (Liu et al., 2019; Sun et al., 2024). The underestimation may stem from either the model or our anthropogenic emission inventory. Observational studies have shown that the decline in anthropogenic emissions is the key driver behind the decrease in $Hg^0$ concentrations at both background sites (Changbai, Ailao, Damei, Waliguan, Chongming) (Feng et al., 2024; Tang et al., 2018), and urban sites (Nanjing) (Sun et al., 2024). To explore reasons for simulation underestimation, we compared the decline rates of observed $Hg^0$ concentration, simulated $Hg^0$ concentration and anthropogenic emissions at these sites, as shown in Table 1. For each site, the decline rate of observed $Hg^0$ concentration was calculated as the difference between maximum value and the concentration at the end of observation period, divided by the

maximum value (see Equation S9 for an example calculating at Changbai). The same method was applied to calculate decline rates for simulated $Hg^0$ concentration, national total $Hg^0$ emissions, $Hg^0$ emissions from the 9 surrounding grids (approximately 500 km × 500 km), and $Hg^0$ emissions from the current grid over the same period.

As shown in Table 1, the decline rates of observed $Hg^0$ concentrations vary across different site types based on their location and emission impacts: (1) Background sites (Changbai, Ailao, Damei, Waliguan): These high-altitude sites with minimal local emissions represent national even global impacts. Their observed $Hg^0$ concentration decline rates closely align with the national total $Hg^0$ emission decline rates and are significantly higher than simulated $Hg^0$ concentration decline rates. (2) Regional background sites (Chongming, Miyun): Located in suburban areas, these sites reflect regional impacts. Their observed $Hg^0$ concentration decline rates align more closely with the emission decline rates from nearby grids (9 surrounding grids) and are also much higher than simulated $Hg^0$ decline rates. (3) Urban sites (Nanjing, Tsinghua, Hohhot): Urban sites are influenced by diverse emission sources, making it difficult to directly associate observed $Hg^0$ concentrations with specific emission types. At Nanjing site, impacted by point source emissions from CFPP and CEM within the local grid, the observed decline rates closely align with local emission decline rates and are higher than simulated rates. At Tsinghua site, impacted by transported emissions from adjacent provinces, the observed $Hg^0$ decline rates are comparable to the national total $Hg^0$ emission decline rates. At Hohhot site, situated at a high altitude and impacted by broader area emissions, the observed $Hg^0$ decline rates align with national total $Hg^0$ emission decline rates.

The observed decline rate matches the emission decline rate and exceeds the simulated rate at all sites. This suggests that our anthropogenic emissions inventory is reasonable and should have reproduced the observed trends. Potential reasons for the model's underestimation include: (1) Boundary conditions. Boundary conditions play a critical role in determining the global background concentration of $Hg^0$ in nested simulations. However, global anthropogenic emissions used in simulations often fail to capture the observed decline trend in $Hg^0$ concentrations. For example, observations from the Northern Hemisphere indicate a decline of approximately 0.011 ng m-3 yr-1, while simulations show only a slight decline of 0.0014 ng m-3 yr-1 (Feinberg et al., 2024). This discrepancy introduces bias in nested simulation trends, particularly at background sites. The inability of boundary conditions to reflect observed trends highlights a key limitation in current simulation. (2) Legacy re-emissions. Legacy re-emissions refer to the re-emission of previously deposited Hg. These $Hg^0$ emissions diffuse back into the atmosphere and are reported to contribute significantly to current atmospheric mercury concentration (Angot et al., 2021) or deposition (Amos et al., 2013). For example, studies suggest that legacy re-emissions account for approximately 60% of atmospheric deposition, compared to 27% from anthropogenic emissions (Amos et al., 2013). (3) Transport process and wind field. Transport process plays a critical role in controlling $Hg^0$ concentrations and trends (Roy et al., 2023), with wind field being a key factor in determining transport process (Brasseur and Jacob, 2017; Yang et al., 2024). By comparing simulated 10 m wind speed from MERRA2 with observed wind speed, we found discrepancies in the monthly wind speed trends between MERRA2 and meteorological observations (Fig. S6). These inconsistencies in monthly trends suggest a potential bias in MERRA2 wind speed data, consistent with findings from other evaluation studies (Miao et al., 2020). Similar biases are observed in wind direction when

comparing MERRA2 with observations (Fig. S7). These biases likely contribute to transport simulation errors and may significantly underestimate $Hg^0$ concentrations in the model.

[Figure]

Figure 5 Temporal and spatial distribution of simulated $Hg^0$ concentration (ng/m$^3$).

Table 1 Decline rate of observed $Hg^0$ concentration, $Hg^0$ emissions, and simulated $Hg^0$ concentration

| Sites | Altitude (m a.s.l.) | Type | Period | Decline rate | | | | |
|---|---|---|---|---|---|---|---|---|
| | | | | Observed $Hg^0$ concentration | Simulated $Hg^0$ concentration | National total $Hg^0$ emissions | $Hg^0$ emission of surrounding 9 grids | $Hg^0$ emission of current grid |
| Changbai | 741 | Background | 2013-2021 | 0.22 | 0.04 | 0.30 | 0.58 | 0.58 |
| Ailao | 2450 | Background | 2012-2021 | 0.42 | 0.03 | 0.35 | 0.12 | 0.09 |
| Damie | 550 | Background | 2012-2021 | 0.46 | 0.25 | 0.35 | 0.38 | -0.14 |
| Waliguan | 3816 | Background | 2013-2021 | 0.29 | -0.02 | 0.30 | 0.13 | -0.12 |
| Chongming | 10 | Regional Background | 2010-2021 | 0.46 | 0.21 | 0.36 | 0.69 | -0.36 |
| Miyun | 128 | Regional Background | 2010-2016 | 0.31 | 0.08 | 0.17 | 0.42 | 0.36 |
| Nanjing | 10 | Urban | 2017-2021 | 0.37 | 0.19 | 0.15 | 0.17 | 0.35 |
| Tsinghua | 50 | Urban | 2015-2021 | 0.32 | 0.06 | 0.26 | 0.29 | 0.38 |
| Hohhot | 1100 | Urban | 2017-2021 | 0.32 | 0.05 | 0.15 | 0.15 | 0.04 |

"

**See revised Manuscript, Lines 308-361.**

2. Extended Validation Metrics: While the model evaluation provides normalized mean bias (NMB) and normalized mean error (NME) for 2021, these metrics alone may not fully capture the model's performance. I suggest incorporating additional metrics, such as the root mean square error (RMSE) and the correlation coefficient (R), to provide a more comprehensive evaluation. Moreover, the evaluation should be conducted seasonally and include spatial analysis to account for variations in mercury emissions throughout the year. This will help ensure the model's performance across different geographic regions and emission sources.

**Response**: We appreciate the reviewer's comments. We have added RMSE and R to evaluate model performance for both only proxy-based and P-CAME at 9 sites in 2020, as this year shows less bias based on the spatial evaluation (Figure 5). Seasonal simulations and observations were compared, and corresponding NMB, NME, RMSE, and R were calculated for each site, as shown in Figure 6.

"3.6 Simulation comparison using P-CAME and only proxy-based inventory

We selected 2020 to compare the simulation differences between the P-CAME and only proxy-based inventories, as 2020 exhibits less bias according to Fig. 5. For each site, we compared seasonal average $Hg^0$ concentrations and evaluated performance using NMB, NME, RMSE, and R, as detailed in Fig. 6. Our analysis revealed that P-CAME have the potentiality to improve simulation accuracy for urban sites, such as Nanjing and Hohhot. In Nanjing site, the grid containing the Nanjing site includes CFPP and CEM point sources. The only proxy-based method underestimates emissions compared to P-CAME (Fig. S8), resulting in lower simulated $Hg^0$ concentrations. P-CAME reduces simulation bias, yielding lower NMB, NME, and RMSE values, indicating better agreement with observations. In Hohhot site, the only proxy-based method tends to overestimate emissions due to the high population density (Fig. S8). By contrast, P-CAME produces lower simulated $Hg^0$ concentrations, which better align with observations, with lower NMB, NME, and RMSE values. These two sites highlight two common scenarios: (1) overestimated emissions in densely populated areas and (2) underestimated emissions in industrial clusters, as discussed in Section 3.1. From this perspective, P-CAME has the capacity to reduce simulation bias by more accurately allocating spatial emissions in urban regions. However, this capacity is currently limited by model bias, such as poor performance in simulating transport processes, as discussed in Section 3.5. For urban sites like Qingdao and Tsinghua, seasonal trends are influenced by air mass sources from different directions, driven by air pressure changes between land and ocean (Shao et al., 2022; Wang et al., 2021). For example, we found that the wind field from MERRA2 does not closely match observations (Fig. S7), which could lead to simulation bias. Since the model struggles to accurately capture these transport processes, its performance at these sites is poor, making it more challenging to identify improvements from revising the emission inventory. The model performs relatively better at rural sites when compared with observations. At these locations, there is little difference in simulation outcomes between using P-CAME and the only proxy-based inventory.

[Figure]

Figure 5 Temporal and spatial distribution of simulated Hg$^0$ concentration (ng/m$^3$).

[Figure]

Figure 6 Comparison of observed and simulated atmospheric mercury concentrations using only proxy-based and P-CAME inventory.

''

**See revised Manuscript, Lines 358-359 and Lines 362-384.**

3. Definition of GEM (L 247): On line 247, the manuscript introduces "GEM" without defining it. While GEM is a well-known term in mercury studies, it is important to spell out "Gaseous Elemental Mercury (GEM)" upon first use to ensure clarity, especially for readers less familiar with the topic.

**Response**: Thank you for your suggestion. We have replaced all instances of GEM with Hg$^0$

and defined Hg$^0$ (Gaseous elemental mercury) upon its first use.

"P-CAME also demonstrates consistency with observed gaseous elemental mercury (Hg$^0$) concentration trends over the past decade"

**See revised Manuscript, Lines 28-29.**

4. Improving Figure 4: In Figure 4, the authors use a bar plot to compare observed and modeled GEM concentrations. While this provides some insight, a range plot (mean with standard deviation) or a box-and-whisker plot would be a better way to represent the variability in the data. Furthermore, a scatter plot could be added to show the correlation between observed and modeled data points, helping readers assess whether the model accurately captures the distribution of GEM concentrations.

**Response**: Thank you for your suggestion. We have replaced the bar plot with a range plot and included NMB, NME, RMSE, and R in the figure to better illustrate the data variability.

"

[Figure]

Figure 6 Comparison of observed and simulated atmospheric mercury concentrations using only proxy-based and P-CAME inventory.

"

**See revised Manuscript, Lines 382-384.**

5. Clarification of Data Availability (L 294): The manuscript mentions the availability of annual mercury emission inventories on figshare. However, the data in figshare (1978, 1980, 1985, etc.) do not appear to match the continuous data shown in Figure 3. It is important to clarify whether "all annual data from 1978 to 2021" will be made publicly available. If only certain years will be shared, this should be clearly stated in both the manuscript and the data repository to avoid confusion.

**Response**: Thank you for your comment. We have uploaded all annual data from 1978 to 2021 to figshare.

"Integrating point source emission inventory (P-CAME) can be accessed from http://doi.org/10.6084/m9.figshare.26076907 (Cui et al., 2024)."

**See revised Manuscript, Lines 386-387 and figshare link.**